# Integrative Methylome and Transcriptome Analysis Reveals Epigenetic Regulation of Pigmentation in Oujiang Color Common Carp

**DOI:** 10.3390/ijms262010001

**Published:** 2025-10-14

**Authors:** Wenqi Zhao, Xiaowen Chen, Ayesha Arif, Zhaoyang Guo, Nusrat Hasan Kanika, Yuehan Song, Jun Wang, Chenghui Wang

**Affiliations:** Key Laboratory of Freshwater Aquatic Genetic Resources Certificated by the Ministry of Agriculture and Rural Affairs, National Demonstration Center for Experimental Fisheries Science Education, Shanghai Engineering Research Center of Aquaculture, Shanghai Ocean University, Shanghai 201306, China

**Keywords:** MBD-seq, pigmentation, methylation, epigenetic regulation, spot formation

## Abstract

Oujiang color common carp display four striking varieties of pigmentation, but their epigenetic basis is unclear. We integrated genome-wide DNA methylation (MBD-seq) and transcriptomes (RNA-seq) from dorsal skin of four Oujiang color common carp varieties with three biological replicates. Black-spotted groups (RB, WB) showed approximately 6% higher global methylation than non-black-spotted groups (WR, WW), with differential methylation enriched in introns (>23%) and intergenic regions (>47%). Integrative analyses revealed a strong inverse association between promoter methylation and gene expression; 96 pigmentation-related genes were identified, spotlighting genes such as *ASIP* and *frmA* as key epigenetically silenced regulators in black-spotted carp. RT-qPCR confirmed directional concordance with RNA-seq for *ASIP*, *frmA*, *DGAT2*, *SCARB1*, and *FOSB*. Pathway enrichment implicated melanogenesis metabolism, tyrosine metabolism, lipid metabolism, and fatty acid metabolism, suggesting an interplay between pigment deposition and metabolic regulation. Collectively, the findings present an exploratory view of epigenetic control of coloration and underscore promoter methylation as a core layer influencing color diversity in Oujiang color common carp.

## 1. Introduction

Coloration is a fundamental biological trait that plays critical roles in species survival, communication, and reproduction, and its evolution is shaped by both genetic and epigenetic factors [1]. In teleost fish, the remarkable diversity of pigmentation patterns driven by various types of chromatophores provides an excellent system for investigating the molecular mechanisms of phenotypic variation. Six major types of pigment cells, including melanophores (black or brown), xanthophores (yellow), erythrophores (red or orange), iridophores (silver, blue, or green), leucophores (white), and cyanophores (blue), have been identified in fish skin, forming complex color patterns through their distribution and structural arrangement [2]. The formation of fish body color is regulated by a complex interplay of genetic, epigenetic, and environmental factors. Understanding the molecular basis underlying pigmentation is therefore crucial not only for developmental biology and evolutionary studies but also for aquaculture, where body color significantly affects commercial value [3,4]. Systematic analysis of the genetic and epigenetic bases of pigmentation elucidates phenotypic plasticity and environmental adaptation, identifies measurable targets for optimizing feed, health evaluation, and molecular breeding, and thereby improves color uniformity and stability [5].

Recent research has focused on elucidating the molecular and genetic drivers of pigment deposition and pattern formation [6]. Specific pigmentation genes and their regulatory networks have been extensively studied, including transcription factors, enzymes, and signaling molecules that regulate the development, differentiation, and pigment synthesis of chromatophores. Notably, DNA methylation has emerged as a central epigenetic mechanism regulating pigmentation, providing both heritable stability and environmental responsiveness [7].

With the development of high-throughput sequencing technologies, various sequencing approaches such as whole-genome bisulfite sequencing (WGBS-seq) and methyl-CpG-binding domain sequencing (MBD-seq) have been utilized to study DNA methylation and reveal epigenetic modifications. For instance, methylation alterations have been implicated in the color phenotype of barramundi (*Lates calcarifer*) [8] and in the differential expression of pigmentation genes in red and black crucian carp [9]. In Leopard coral grouper (*Plectropomus leopardus*) [10], genes such as *pmel*, *acsl4*, and *elovl7* have been identified as negatively regulated by DNA methylation. In *Astatotilapia burtoni* [11], increased partial methylation levels of the *ednrb* promoter were found to be associated with the development of yellow coloration on the body surface. These examples demonstrate that DNA methylation is a key molecular mechanism regulating pigmentation formation. However, research in this area remains limited, with a lack of systematic multi-omics studies integrating DNA methylation and transcriptomic dynamics.

The Oujiang color common carp (*Cyprinus carpio* var. *color.*), a culturally significant breed with over 1200 years of cultivation history in China, exhibits four distinct body color varieties: pure white (WW), pure red (WR), red with black patches (RB), and white with black patches (WB) [12] derived from three base colors (black, red, and white). These varieties segregate within the same full-sib family, which enables comparisons under a controlled genetic background. Early population and breeding work established the heritable nature of the color traits and the practical value of selective breeding, and genomic resources for common carp and related cyprinids have continued to expand [13,14]. Molecular studies in Oujiang colored common carp show that differences in skin coloration are accompanied by changes in gene expression, with repeated enrichment of melanogenesis, tyrosine metabolism, and lipid or energy metabolism, especially in comparisons between red and white varieties [15,16]. Despite these advances, the epigenetic basis of color divergence in this lineage remains incompletely resolved because most prior studies examined expression or methylation in isolation. This gap motivates an integration of the methylome and the transcriptome in a single pedigreed lineage to map morph-specific methylation landscapes.

The objective of this study was to elucidate the epigenetic and transcriptional mechanisms underlying pigmentation diversity in Oujiang color carp. We generated genome-wide DNA methylation maps, identified differentially methylated regions (DMRs), and characterized transcriptomic profiles to detect differentially expressed genes (DEGs) associated with pigmentation. Finally, we integrated methylome and transcriptome data to examine the regulatory relationship between DNA methylation and gene expression.

## 2. Results

### 2.1. Whole Genome Methylation Profiles Among Four Common Color Carp Varieties

Genome-wide analysis revealed distinct methylation patterns, with black-spotted varieties (RB and WB) exhibiting higher overall methylation coverage than non-spotted varieties (WR and WW) (Figure 1a). All four-color varieties exhibited typical “valley-like” methylation patterns around transcription start sites (TSSs) and transcription end sites (TESs). The genomic DNA methylation levels were lower in the regions 2 kb upstream and downstream of genes across all groups, with relatively higher methylation levels identified at TSS and TES regions (Figure 1b). A total of 2,1176, 2,1517, 1,6467, and 2,0857 methpeaks (methylation peaks) were identified in RB, WB, WR, and WW, respectively, with the black-spotted group (RB and WB) showing higher peak numbers than the non-spotted group (WR and WW), and WB displaying the highest overall (Figure 1c). Despite differences in peak counts, the overall distribution patterns of methylation peaks across genomic features were largely similar among all varieties (Figure 1d). Annotation revealed enrichment of methylation peaks in intronic and intergenic regions, with moderate representation in promoter regions (Figure 1d). Notably, the WW variety displayed a significantly higher proportion of methylation peaks distributed in promoter regions compared to the other varieties (*p* < 0.01), suggesting that its gene expression may be subject to strong promoter methylation regulation (Figure 1d). In general, the black-spotted varieties (RB and WB) showed more methylation peaks, higher methylated genome coverage, and elevated global methylation levels compared to non-black-spotted varieties (WR and WW) (Figure 1).

### 2.2. Differential Methylation Analysis Among the Four-Color Varieties

After quality control of the raw data, the proportion of bases with Q30 scores exceeded 89.76% for all libraries, yielding 39, 32, 31, and 23 million clean reads, respectively (Appendix A). A total of 1331, 1649, 365, and 951 differentially methylated regions (DMRs) were identified between RB and WB, WB and WW, WR and WW, and RB and WR, respectively (Appendix A). Chromosomal distribution analysis revealed that DMRs in all four pairwise comparisons were predominantly localized to specific chromosomes (Figure 2a). Notably, comparison groups exhibited distinct distribution patterns (Figure 2a). Black-spotted vs. non-black-spotted comparisons (RB vs. WR and WB vs. WW) showed similar DMR chromosome distributions (Figure 2a). The obtained DMRs could be classified into hypermethylated and hypomethylated regions across the genome. In comparisons of RB vs. WR, WB vs. WW, RB vs. WB, and WR vs. WW, we identified 354, 820, 893, and 227 hypermethylated regions, respectively, along with 595, 827, 438, and 139 hypomethylated regions (Figure 2b). Comparative analysis revealed the highest number of hypermethylated regions in RB vs. WB and the lowest in WR vs. WW, while hypomethylated regions were most abundant in WB vs. WW and least in WR vs. WW (Figure 2b).

CpG context analysis showed methylated regions enriched in CpG shores and shelves, with relatively fewer DMRs in CpG islands (Figure 2c). Compared to the genomic background group, the distribution of DMRs within CpG islands and their adjacent areas showed no significant changes (Figure 2c). This pattern aligns with a common epigenetic feature in eukaryotes, where functional DNA methylation is often enriched around, rather than within, CpG islands. More than 50% of the DMRs were in intergenic and intronic regions, while only 5% were situated in promoter and exon regions (Figure 2c). This suggests that epigenetic regulation may be mediated by distal regulatory elements in common carp. The distribution of hyper- and hypomethylation displayed distinct patterns associated with body color. In both black-spotted comparisons (RB vs. WR; WB vs. WW), non-black-spotted groups (WR and WW) exhibited more hypermethylated regions than their black-spotted counterparts (RB and WB) (Figure 2d). Similarly, in red–white comparisons, red-colored groups (RB and WR) showed a greater number of hypermethylated regions than white-colored groups (WB and WW) (Figure 2d).

We identified the corresponding differentially methylated genes (DMGs) associated with these DMRs (Appendix A). GO (Gene Ontology) enrichment analysis revealed that the DMGs were primarily enriched in pathways such as “G protein-coupled receptor signaling pathway”, “neural crest cell fate commitment”, and a series of other pathways (Appendix A).

### 2.3. Different Expression Profiles of the Four-Color Varieties

We obtained 27, 25, 29, and 24 million clean reads from the skin tissues of RB, WB, WW, and WR, respectively (Appendix A). A total of 2023, 1911, 1078, and 937 DEGs were identified between the RB and WB, WB and WW, WR and WW, and RB and WR comparison groups (Figure 3a,b). Among the 96 pigmentation-related genes, such as tyrosinase-related protein 1 (*TYRP1*) and scavenger receptor class B member 1 (*SCARB1*), differentially expressed (Appendix A). In the comparison between groups with and without black spots, we observed downregulation of several pigmentation-related DEGs, including *ASIP* and *frmA*. Likewise, in the red versus white comparison, genes such as *SCARB1* and *SREBF1* were upregulated (Figure 3a).

In the black-spotted versus non-black-spotted comparisons, the Venn analysis shows 114 shared DEGs between RBvsWR and WBvsWW, with 823 and 1909 DEGs unique to each contrast, respectively (Figure 3b). For the red-versus-white comparisons, RB vs. WB and WR vs. WW share 272 DEGs, with 811 and 1644 unique to each, respectively (Figure 3b). Across all four contrasts, only five DEGs are shared (Figure 3b).

Functional enrichment analyses were performed on DEGs from each of the four comparison groups (Figure 3). Enrichment analysis of DEGs revealed comparison-specific pathway associations. For the black-spotted vs. non-spotted (RB vs. WR, WB vs. WW) groups, DEGs were significantly enriched in pigment- and melanin-related pathway. Among these terms, we identified genes previously reported to be associated with body coloration, such as *PMEL*, *EDNRB*, and *TYRP1* (Figure 3c). Similarly, in the comparison between red and white skin varieties (RB vs. WB, WW vs. WR), GO enrichment analysis of DEGs revealed significant enrichment in lipid metabolism and pigmentation-related pathways (Appendix A). KEGG (Kyoto Encyclopedia of Genes and Genomes) pathway analysis further indicated that these genes were mainly involved in fatty acid metabolic processes (Figure 3c). A heatmap of DEGs revealed that individuals with black-spotted phenotypes (RB and WB) exhibited similar expression patterns, while individuals without black spots (WR and WW) also showed similar patterns (Figure 3d). Moreover, opposite expression trends were observed between the black-spotted and non-spotted groups (Figure 3d). Gene-wise clustering further grouped the differentially expressed genes into two major categories: lipid metabolism-related genes and pigmentation-related genes (Figure 3d). Among the pigmentation-related genes, the black-spotted groups RB and WB displayed consistent expression trends, whereas in the lipid metabolism-related genes, they exhibited opposite expression patterns (Figure 3d). A similar trend was observed in the non-spotted groups WR and WW (Figure 3d). These findings are consistent with the previous functional enrichment results, further supporting the hypothesis that these genes may play important roles in body coloration, particularly in the molecular regulation of black spot formation (Figure 3d).

To validate the transcriptomic results, we quantified five candidate genes (*ASIP*, *frmA*, *DGAT2*, *SCARB1*, and *FOSB*) by RT-qPCR across four color varieties (RB, WB, WW, and WR) and compared the expression trends with RNA-seq results (Figure 4). The two methods showed highly concordant intergroup changes: *ASIP* was highest in WW and lowest in RB, with WR intermediate; *frmA* likewise peaked in WW, with lower levels in RB/WB and intermediate expression in WR, *DGAT2* increased progressively from RB, WB, WW, and WR and was highest in WR; the expression of *SCARB1* was higher in RB, WR and was lower in WB, WW; and *FOSB* was highest in WB and decreased toward WR. Overall, the RT-qPCR data closely recapitulated the RNA-seq differences among groups, supporting the reliability of the transcriptomic quantification and revealing morph-specific expression patterns associated with pigmentation (Figure 4). The RT-qPCR primer sequences. Using *β-actin* as the reference, standard curves were generated for *ASIP*, *frmA*, *DGAT2*, *SCARB1*, and *FOSB*. The coefficients of determination were R^2^ = 0.946–0.999, and amplification efficiencies were E = 0.91–1.07 (≈91–107%), all within the commonly accepted range for quantitative analysis (~90–110%) (Appendix A).

### 2.4. Integrated Transcriptome-Methylome Analysis

By integrating the transcriptome and methylome results, 30, 38, 17, and 5 genes that exhibit both differential methylation and differential expression were identified (Figure 5a). Analysis of the overlap between DMRs and DEGs revealed that a significantly higher level of DEGs (15–31%) overlapped with DMRs than the background genes (1.6%; *p* < 0.01). This suggests that a large portion of differentially expressed genes may be regulated by DNA methylation (Figure 5b).

Analysis of methylated regions spanning 2 kb upstream and downstream of genes revealed that genes with low expression levels exhibited higher methylation in promoter regions, suggesting a negative correlation between promoter methylation and gene expression (Figure 5c). Conversely, methylation within the gene body and downstream regions was positively correlated with gene expression (Figure 5c). Furthermore, a subset of negatively correlated genes related to pigmentation were identified, including *ASIP*, *frmA*, *SCARB2*, *RBP2*, *FOXD3*, *MITF*, and *MC5R.* These genes displayed inverse relationships between methylation and expression levels, reinforcing their likely role in methylation-mediated regulation of pigmentation (Figure 5d).

Through comprehensive integration of transcriptomic and epigenomic data, we identified two high-confidence candidate genes, *ASIP* and *frmA,* that likely contributed to body color through promoter methylated regulation (Figure 6). Significant DMRs were present in the promoters of *ASIP* (Chr18:25,220,001–25,220,500) and *frmA* (Chr23:16,493,501–16,494,000). Across the four varieties, both genes exhibited higher promoter methylation in the black-spotted groups (RB, WB) and lower methylation in the non-spot groups (WR, WW) (Figure 6). Consistently, the mRNA abundance (log_2_[TPM + 1]) is reduced in RB/WB and elevated in WR/WW (Figure 6). This pattern was observed in both contrasts (RB vs. WR and WB vs. WW), indicating that promoter hypermethylation is associated with reduced gene expression in the black-spotted fish (Figure 6).

## 3. Discussion

The vibrant coloration patterns in fish underscore the intricate interplay of genetics, epigenetics, and environmental influences. Genetic factors such as coding mutations and transcriptional regulation are major determinants of color traits. However, many studies have also emphasized that DNA methylation plays a significant and adaptive role in pigmentation control [17,18]. Across species, DNA methylation modulates pigmentation by silencing key genes when CpG sites in regulatory regions are hypermethylated, as shown at the Agouti locus in mice [19] and the MITF promoter in quail [20]. In the Yesso scallop [21], differentially methylated regions cluster in pigment biosynthesis pathways and show an inverse relation to gene expression, linking methylation shifts to color change. These studies illustrate that DNA methylation can mediate body coloration by regulating the pigment synthesis pathway. Our genome-wide DNA methylation analysis in Oujiang color common carp revealed distinct methylation profiles among four pigmentation types, highlighting a potential epigenetic basis for body color divergence in teleost’s (Figure 1, Figure 2, Figure 3 and Figure 4).

### 3.1. Regulatory Effects of DNA Methylation on Gene Expression in Oujiang Color Common Carp

DNA methylation regulates gene expression in a context-dependent manner: at CpG-rich promoters it can interfere with transcription-factor binding and is often linked to stable repression, whereas methylation within gene bodies or downstream regions often tracks active transcription and can influence alternative splicing [22]. This context dependence aligns with evidence from the purple sea urchin, where methylation-expression relationships vary with promoter accessibility [23]. In our Oujiang color common carp dataset, promoter methylation showed a broadly negative correlation with gene expression, while gene-body and downstream methylation showed a positive correlation, supporting a regulatory role for DNA methylation in this species (Figure 4).

### 3.2. Epigenetic Differences Between Black-Spotted and Non-Black-Spotted Color Common Carp

Distinct methylation and gene expression profiles between black-spotted (RB, WB) and non-spotted (WR, WW) groups suggest that melanin patterning in Oujiang color carp may also be epigenetically modulated. The clustering of DEGs and DMGs shows consistent grouping based on the presence of black pigmentation (Figure 1, Figure 2 and Figure 3), indicating shared regulatory mechanisms. Enrichment of DMRs and DEGs in pathways such as melanogenesis, GPCR signaling, and lipid metabolism aligns with the functional role of chromatophores and their metabolic demands during pigment production. In one study [24], methylation profiling during the differentiation of zebrafish neural crest cells into melanocytes revealed that most promoter regions did not undergo significant epigenetic changes. Instead, that study highlighted the cis-regulatory role of distal enhancers in melanocyte differentiation. This finding is consistent with our observation that DMRs were predominantly located in intronic and intergenic regions (Figure 2). Furthermore, in our study, genes associated with DMRs were significantly enriched in pathways related to “neural crest cell fate determination”, supporting the idea that DNA methylation may influence melanocyte formation and, consequently, the development of black pigmentation. These results indicate that DNA methylation may influence melanocyte formation and contribute to black spot development in Oujiang color carp.

### 3.3. Epigenetic Differences Between Red and White Related Color Common Carp

Lipid metabolism is a key driver of body color formation in teleost fish. Studies have shown that red or carotenoid-rich skin regions are closely associated with the upregulation of lipid metabolism genes (such as *SCARB1*, *DGAT2*, and *plin6*), which are involved in carotenoid transport, esterification, and lipid droplet aggregation [10,25,26]. In an East African cichlid (*Tropheus duboisi*), carotenoid-rich yellow skin shows higher expression of *SCARB1*, *DGAT2*, and the teleost-specific perilipin *plin6* relative to adjacent white skin, directly linking lipid uptake, esterification, and lipid-droplet biology to integumentary coloration [25]. Beyond cyprinids and cichlids, diet-driven astaxanthin deposition in salmonids also recruits lipid and retinoid metabolic programs during flesh pigmentation, underscoring the conserved involvement of lipid handling in carotenoid processing [27]. In the comparison between the red and white groups, we identified several genes related to lipid metabolism, such as *SCARB1*, *DGAT2*, and *GC1A*, whose expression levels were significantly higher in the red tissue than in the white group. This suggests that lipid metabolism plays an important role in red coloration. Furthermore, we observed varying degrees of methylation in these genes, with methylation levels showing a certain negative correlation with their expression. For example, in the promoter region of the *SCARB1* gene, DNA methylation levels and gene expression show a significant negative correlation. These results are consistent with the findings mentioned above and indicate that DNA methylation may potentially contribute to red coloration.

### 3.4. Pigmentation-Related Genes and Pathways

Genes governing melanosome transport and positioning were prominent. The *RAB27–MLPH–myosin* axis tethers mature melanosomes to cortical actin for peripheral dispersion and transfer [28], and *MYO7A* is essential for melanosome motility in retinal pigment epithelium. Lineage and survival signaling that pattern the melanocyte compartment also shifted; *EDNRB* promotes early expansion of melanocyte precursors, and *KIT* supports survival across development [29]. Consistent with upstream transcriptional control, *PAX3*, a regulator of melanocyte fate that modulates *MITF*, was differentially expressed in several contrasts [30]. Beyond melanin, our data highlight carotenoid and lipid, *SCARB1*, a high-density lipoprotein receptor that mediates cellular carotenoid uptake and is required for carotenoid-based coloration in birds [31], was strongly regulated. Lipid storage and synthesis nodes such as *DGAT2* and *SREBF1* may reshape the lipid milieu that hosts and stabilizes pigments [32]. We also noted coordinated changes in fatty acid catabolism that influence substrate flux: *CPT1A* [33] gates mitochondrial β-oxidation, and *PDK2* [34] inhibits *PDH* to bias carbon toward lipid oxidation. Together these shifts are consistent with metabolic reprogramming that can affect melanogenesis efficiency and carotenoid storage.

In general, the pigment differential genes we identified are mainly located in the melanin pathway and fat metabolism pathway, which are clearly related to the body color of the experimental subjects we selected. These patterns align with our GO and KEGG enrichments and support a model in which body coloration in Oujiang color carp emerges from the joint remodeling of melanin synthesis transport and lipid–carotenoid metabolism, rather than from single-gene effects.

### 3.5. Molecular Mechanisms Underlying Skin Coloration in Oujiang Color Common Carp

Agouti-signaling protein (*ASIP*) acts as an endogenous antagonist of the α-MSH/*MC1R* signaling pathway and has been extensively demonstrated to regulate pigment cell development and melanin deposition [35,36,37]. In zebrafish, *Asip1* is expressed dorsoventrally, suppressing melanophore differentiation ventrally, while loss-of-function mutants show ventral hyperpigmentation [38]. Meanwhile, in mice [39], CpG methylation at an upstream retrotransposon modulates *ASIP* expression, with hypomethylation driving ectopic expression and yellow coats and hypermethylation restoring wild-type pigmentation. In our study, we observed that *ASIP* exhibited low promoter methylation and high expression in non-black-spotted groups (WR, WW), whereas in black-spotted regions, the promoter was more methylated and *ASIP* expression was significantly reduced. Drawing parallels with the above models, we propose the following potential mechanism: In non-black-spotted skin, low promoter methylation leads to higher *ASIP* expression, which antagonizes *MC1R-α-MSH* signaling, suppressing the Wnt/MAPK/MITF axis, reducing melanogenesis, and maintaining a no-black (lighter) phenotype. In black-spotted regions, promoter hypermethylation lowers *ASIP* expression, allowing sustained *MC1R* signaling that activates *MITF*/tyrosinase pathways and drives melanin deposition, producing visible black spots. This mechanism is consistent with the coordinated DMR–DEG patterns we observed, and it offers a unified explanation for the coexistence of black spots. However, it needs molecular evidence to confirm this epigenetic regulation in future study.

The *frmA* gene, orthologous to human *ADH5* (S-(hydroxymethyl)-glutathione dehydrogenase), plays a key role in cellular GSH-dependent detoxification of formaldehyde, catalyzing the oxidation of S-(hydroxymethyl)-glutathione to formate, a reaction that consumes reduced glutathione (GSH) [40]. In mammalian systems, promoter CpG hypermethylation of *ADH* family genes is strongly associated with reduced gene expression, especially in cancer tissues, underscoring the significance of methylation-mediated gene silencing [41]. Intracellular GSH levels serve as a critical redox buffer in melanocytes; sulfhydryl-based antioxidants such as GSH can influence *TYR* (tyrosinase) activity and melanin synthesis pathways, with dysregulated GSH levels impacting melanin production and oxidative balance [42]. Additionally, microsomal glutathione S-transferase 1 (*MGST1*), which requires GSH as a cofactor, has been demonstrated to promote melanin biosynthesis and melanosomal eumelanin accumulation in melanoma models, linking glutathione metabolism to pigment formation [43]. In our dataset, *frmA* displayed low promoter methylation and high expression in non-black-spotted fish (WR and WW) (Figure 5). In contrast, in black-spotted regions, the promoter of *frmA* was hypermethylated, coinciding with a significant reduction in gene expression. We hypothesize that reduced *frmA* expression may lead to intracellular GSH accumulation, which indirectly enhances eumelanin synthesis and contributes to black spot formation. However, further molecular experiments are needed to confirm this hypothesis.

### 3.6. Translational Relevance: Methylation Markers for Selective Breeding

This study integrates MBD-seq and RNA-seq to reveal methylation–expression coupling in the melanogenesis and lipid–carotenoid pathways underlying body color formation in Oujiang color common carp. We observed directionally consistent changes at loci such as *SCARB1*, *DGAT2*, *ASIP*, and *frmA* that align with the phenotypes, indicating that methylation marks near these genes have potential as auxiliary indicators for selective breeding. Our results provide candidate sites and a practical path for applying methylation markers in the selection of ornamental koi and aquaculture carp. Specifically, we identified stable DMRs near pigmentation and lipid-metabolism genes that show concordant associations with transcription, and these sites can be converted into targeted methylation assays for germplasm grading and broodstock screening. Methylation metrics can also be incorporated as covariates alongside genome-wide variant information to add an epigenetic layer of decision support in breeding programs [44,45].

## 4. Materials and Methods

### 4.1. Sample Collection

This study was approved by the Institutional Animal Care and Use Committee (IACUC) of Shanghai Ocean University (Shanghai, China) (approval number: SHOU-DW-2021-018). All sampling procedures complied with the IACUC guidelines on the care and use of animals for scientific purposes. The study was conducted on specimens of Oujiang color carp with four contrasting color patterns (RB, WB, WW, and WR) from a full-sibling family (RB♀ × WW♂). Three individuals from each color type were selected, and fresh dorsal skin samples (red/black from RB, white/black from WB, white from WW, and red from WR; 0.8–1 cm^2^) were collected and immediately preserved in liquid nitrogen for subsequent.

### 4.2. DNA and RNA Extraction

DNA and RNA were extracted from skin samples using a DNA/RNA extraction kit (Qiagen, Hilden, Germany) following the manufacturer’s protocol. Reverse transcription (RT) was performed using the Superscript III FS cDNA Synthesis SuperMix kit (Life Technologies, Carlsbad, CA, USA) to synthesize total cDNA [46].

### 4.3. MBD-Seq Sequencing

Genomic DNA from skin samples was sonicated using a Bioruptor (Diagenode, Liege, Belgium) to produce fragments of ~100–350 bp. Fragmented DNA was enriched for methylated regions using the MethylMiner™ methylated DNA kit(Invitrogen, Carlsbad, CA, USA.) and purified with a 2000 mM NaCl buffer. The enriched DNA was used to construct libraries for high-throughput MBD-seq analysis on the Illumina HiSeq 2500 platform(Illumina, Inc., San Diego, CA, USA). Three individuals per color type were analyzed [47].

### 4.4. Genome-Wide Methylome Profiling

Quality control of MBD-seq reads was performed using FastQC (v 0.12.0) and fastp (v 0.25.0) [48] to remove low-quality and adapter sequences. Clean reads were aligned to the common carp reference genome using Bowtie2 (v 2.3.5.1) [49] and sorted using SAMtools (v1.5) [50]. Peak detection was performed with MACS3 (v 3.0.1) [51] at a *q*-value threshold of 0.05. Peaks were filtered using Bedtools (v 2.28.0) [52]. Three biological replicates per group were merged. Deeptools (v 2.0) [53] was used to assess read distribution and generate bigWig files for IGV (v 2.8.0) [54] visualization. Methylation peaks were annotated using ChIPseeker [55], and differentially methylated regions (DMRs) were identified using the MEDIPS R package (v 1.43.1) [56]. The genome was divided into 500 bp bins, and bins with adjusted *p* < 0.05 and |log2 fold change| ≥ 0.585 were considered DMRs. DMRs were annotated using the annotate Peaks.pl function in HOMER (v 4.11.1) to identify differentially methylated genes (DMGs).

‘Promoter-TSS’ means extended regions around the transcription start site (TSS) from −1000 bp to +100 bp, while transcription end sites (TESs) indicate extended regions around TTS from −100 bp to +1000 bp. CpG island coordinates were retrieved from the UCSC Genome Browser. The regions flanking CpG islands (±2 kb) were designated as “CpG shores,” while the distal regions (±2–4 kb) were defined as “CpG shelves.” The Genome-wide Methylation Coverage Analysis Using “bedtools genomecov”.

### 4.5. RNA-Seq Sequencing and Data Analysis

The integrity and quantity of RNA were assessed using an Agilent 2100 Bioanalyzer (Agilent, Shanghai, China). A total of 18 RNA samples with RNA integrity number (RIN) > 7.0 were used. The NEB Next^®^ Ultra™ RNA Library Prep Kit(New England Biolabs (NEB), Ipswich, MA, USA.) was utilized to construct RNA-Seq libraries for three biological replicates each from RB, WB, WR, and WW. Sequencing was performed on the Illumina HiSeq 2500 platform, generating 150 bp paired-end reads. The data preparation process was the same as that for MBD-Seq, where RNA-Seq data were preprocessed and aligned to the common carp reference genome. The gene read counts were determined using feature counts, and the raw read counts were combined into a count matrix and used as input for DESeq2 [57]. Differentially expressed genes (DEGs) were identified through DESeq2 analysis, where expression changes were considered significant with |log2(FC)| ≥ 1 and *p*-value < 0.05. Based on coloration, we classified RB and WB as the black-spotted group (presence of black spots), with WR and WW as the no-black-spotted group. In parallel, RB and WR were designated the red body color group, whereas WB and WW constituted the white body color group. Gene function annotation and enrichment analysis were conducted using the ClusterProfiler R package (v3.9), focusing on Gene Ontology (GO) and Kyoto Encyclopedia of Genes and Genomes (KEGG), where a *p*-value < 0.05 was considered significant. From the significantly enriched pathways, we selected DEGs that belong to melanogenesis, tyrosine metabolism, lipid metabolism, fatty acid metabolism, and related different regulatory processes. These DEGs were designated pigmentation-related genes. To increase robustness, we report cross-contrast recurrence (the same gene meeting the criteria in at least two contrasts) and prioritize recurrent genes in the Results section.

### 4.6. RT-qPCR Verification of RNA-Seq Results

To validate the RNA-seq results, five representative genes were assayed by quantitative real-time PCR (RT-qPCR), with β-actin as the internal reference gene. Reactions (20 µL) contained 10 µL Hieff UNICON^®^ qPCR SYBR Green Master Mix(Yeasen Biotechnology (Shanghai) Co., Ltd., Shanghai, China.), 0.4 µL each gene-specific forward and reverse primer, 1 µL cDNA template, and 8.2 µL nuclease-free water. Thermocycling was performed on a real-time PCR thermocycler under the following program: 95 °C for 2 min; 40 cycles of 95 °C for 10 s and 60 °C for 30 s. Relative transcript abundance was calculated using the 2^−ΔΔCt^ method [58] with *β-actin* [59] as the internal control and the corresponding control group as the calibrator. Technical triplicates were run for each sample and target.

### 4.7. Statistical Analysis

Statistical analyses were performed using GraphPad Prism 8.01 (GraphPad Software, Boston, MA, USA) and SPSS 24.0 (IBM Corp., Armonk, NY, USA). Chi-square tests were applied to assess the annotation proportions of DNA methylation peaks across genomic features, while inter-group comparisons of quantitative variables (including gene expression levels) were analyzed by one-way ANOVA, with a *p*-value < 0.05 considered statistically significant.

## 5. Conclusions

We constructed both DNA methylation and transcriptome profiles for Oujiang color common carp with different skin pigmentation patterns, revealing significant differences between black-spotted and non-black-spotted groups at both epigenetic and transcriptional levels. Our analysis further explored the role of DNA methylation in pigmentation, particularly concerning lipid metabolism–related pathways. Additionally, we investigated the potential mechanisms of methylation-mediated transcriptional regulation in Oujiang color common carp. We identified genes, such as *ASIP* and *frmA*, whose expression appears to be regulated by DNA methylation. These findings provide important clues for further study of the epigenetic mechanism of body color regulation.

## Figures and Tables

**Figure 1 ijms-26-10001-f001:**
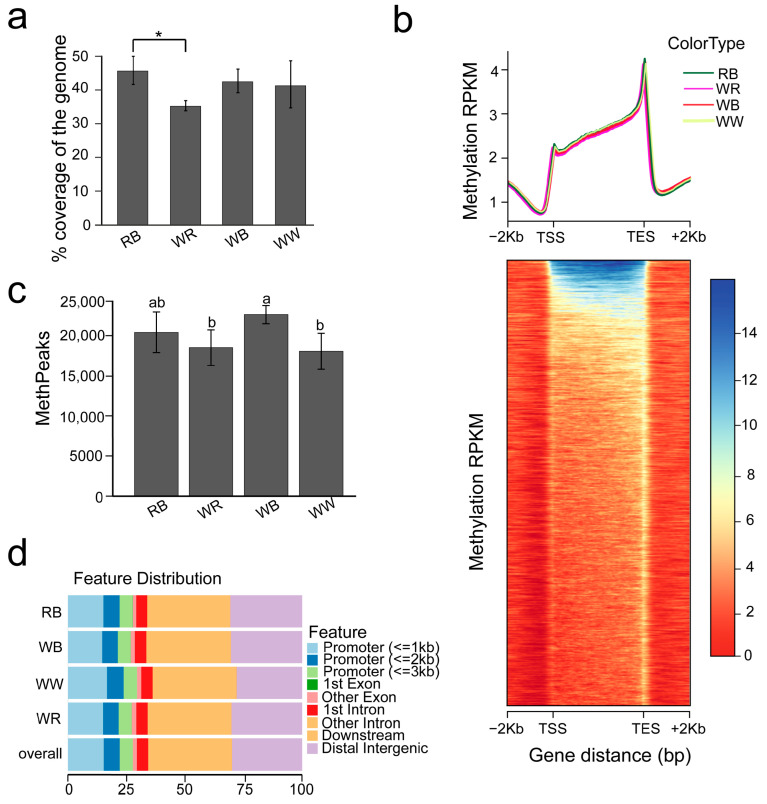
Genome-wide DNA methylation profiling across four pigmentation morphs. (**a**) The bar plot shows genome-wide methylation coverage. (**b**) The distribution of MBD-seq reads in the region around the gene body. (**c**) Bar plots display predicted methylation peaks across pigmentation morphs. (**d**) The distribution of methylation peaks across different genomic features. Asterisks indicate statistical significance between groups (* *p* < 0.05; one-way ANOVA followed by Tukey’s HSD). Letters with different a and b indicate significant difference between groups at *p* < 0.05 (one-way ANOVA followed by Tukey’s HSD; multiple-comparison p values adjusted as specified).

**Figure 2 ijms-26-10001-f002:**
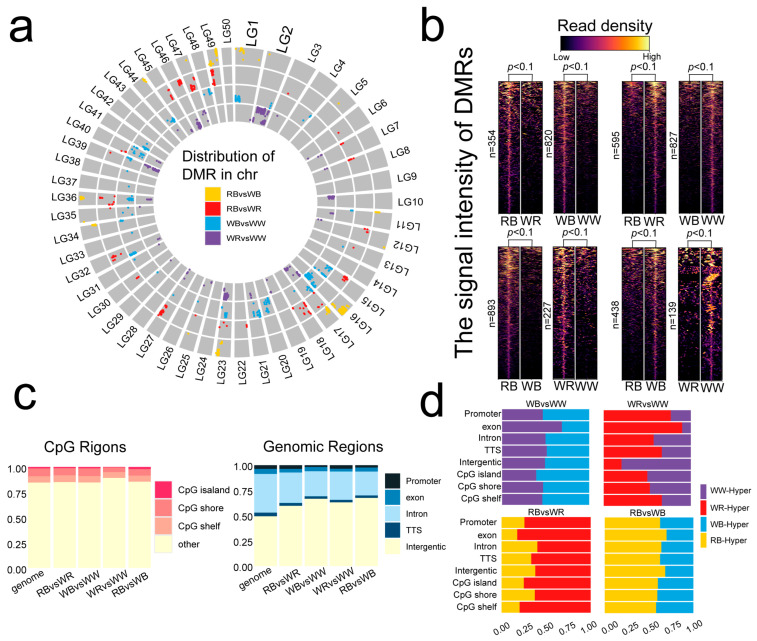
Differential methylation analysis. (**a**) Chromosomal distribution of DMRs identified in four pairwise comparisons. (**b**) DMRs, hypermethylated and hypomethylated in four pigmentation variants. The enrichments shown in the heatmap were calculated within a ±5 kb window around each DMR. (**c**) The distribution of DMRs across CPG and genomic regions. (**d**) Bar plots showing the proportion of hypermethylated regions across genomic features.

**Figure 3 ijms-26-10001-f003:**
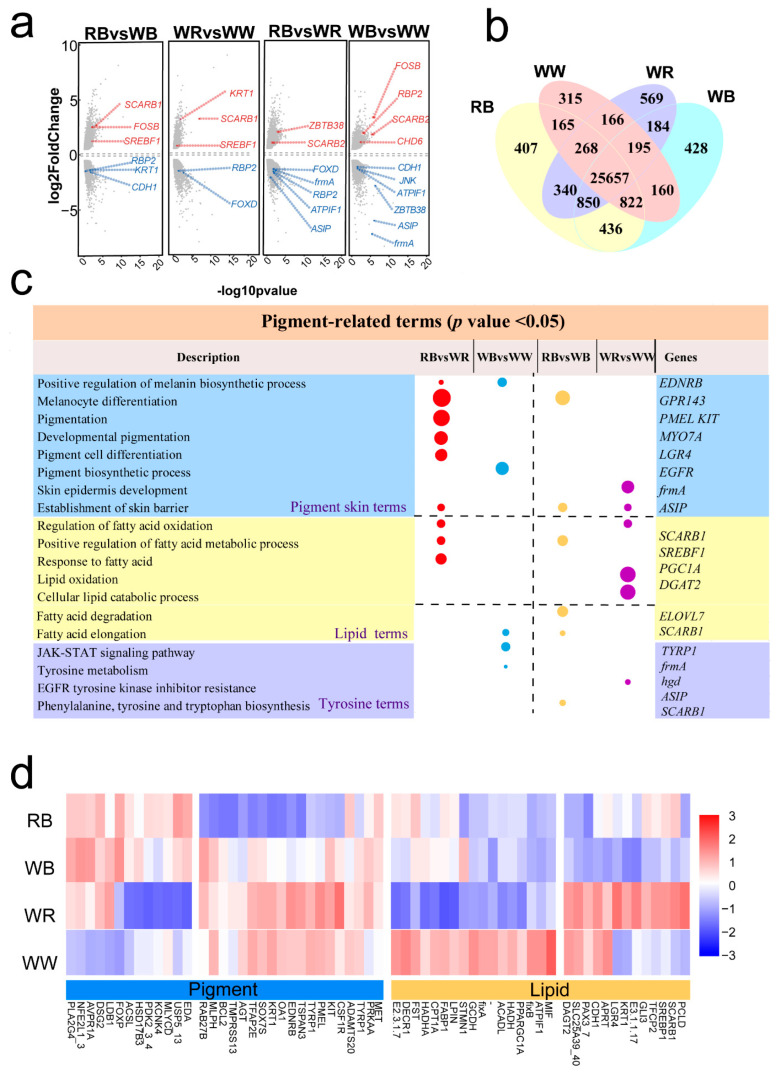
Differential gene expression analysis between groups with different colors. (**a**) The volcano plot shows the differentially expressed genes screened out through transcriptome analysis, and some genes related to body color are listed on the right side. (**b**) Venn diagram showing the overlap of DEGs among the four pairwise comparison groups. (**c**) Enriched pathways of DEGs in each of the four comparisons. (**d**) Heatmap showing the expression patterns of pigmentation-related genes across the four varieties.

**Figure 4 ijms-26-10001-f004:**
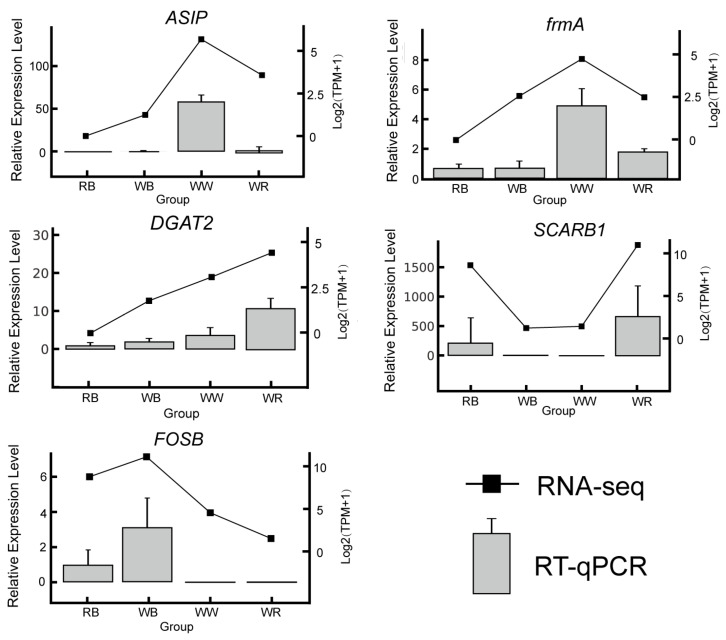
The relative expression levels of 5 DEGs were verified by RT-qPCR compared with RNA-seq, respectively. With β-actin as an internal reference gene. gray bars, qRT-PCR; black line, RNA-seq.

**Figure 5 ijms-26-10001-f005:**
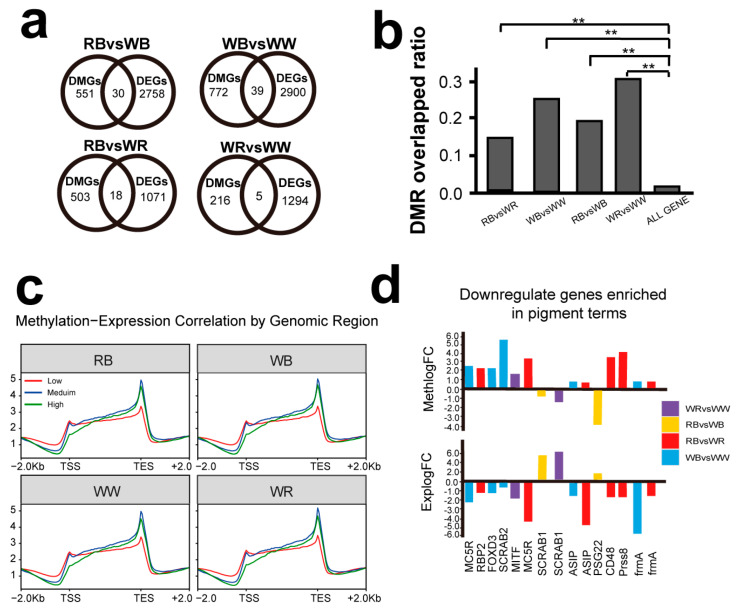
Integrated transcriptome methylome analysis. (**a**) The venn diagram of the numbers of overlapping DEGs-DMGs in the four group comparisons. (**b**) DMR overlapped ratio in the four groups of comparisons. (**c**) The correlation of methylation level with expression in different varieties. (**d**) Pigmentation terms associated genes epigenetically downregulated via methylation. Asterisks indicate statistical significance between groups (**: *p* < 0.01; one-way ANOVA followed by Tukey’s HSD).

**Figure 6 ijms-26-10001-f006:**
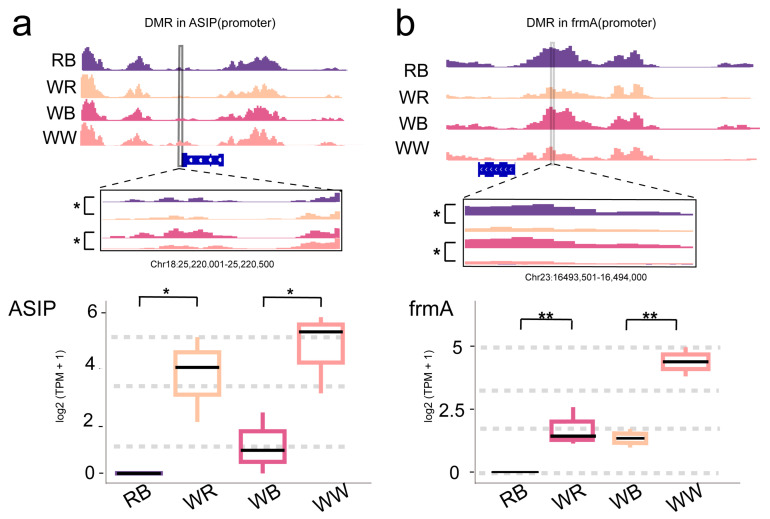
Candidate gene methylation and expression associated with body color. (**a**) Methylation peak profile of *ASIP* promoter across four varieties, with boxplot showing gene expression. (**b**) Methylation peak profile of the *frmA* promoter and boxplots showing gene expression in four varieties. Asterisks indicate statistical significance between groups (*: *p* < 0.05; **: *p* < 0.01; one-way ANOVA followed by Tukey’s HSD).

## Data Availability

The RNA-seq and MBD-seq datasets have been deposited in the Genome Sequence Archive (GSA) at CNCB-NGDC under accession numbers CRA025590 and CRA030041, available at https://bigd.big.ac.cn/gsa/browse/CRA025590 (accessed on 11 October 2025) and https://bigd.big.ac.cn/gsa/browse/CRA030041 (accessed on 11 October 2025).

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
