# Peer review of "Integrative Methylome and Transcriptome Analysis Reveals Epigenetic Regulation of Pigmentation in Oujiang Color Common Carp"

_ijms, 2025, doi:10.3390/ijms262010001_

Round 1
Reviewer 1 Report
Comments and Suggestions for Authors
In the present study, authors revealed epigenetic regulation of pigmentation in Oujiang color carp through integrative methylome and transcriptome analysis. However, the sample number for each color type is too less. Only three individuals were collected for each color type. The lack of biological replicates renders the findings of this study unreliable. Thus, I suggested rejection at the present form. Authors should collect more individuals for each color type, and prepared more biological replicates.
In abstract, authors should briefly describe the results in the present study.
In introduction, describe more related to the oujiang common carp. In addition, please briefly describe the reason why it is needed to analyse the molecular mechanism of pigmentation.
In section 2.3, how authors selected the pigmentation-related genes? It is not clear. Please re-state this section.
The expressions of DEGs should be verified by qPCR.
In discussion section, it is suggested to introduce the functions of pigmentation-related pathways and genes, identified in the present study.
Three individuals from each color type were selected. The sample number is to less. At least, three individuals were pooled together to form a biological replicated, and at least three biological replicates should be prepared for each color.
Reviewer 2 Report
Comments and Suggestions for Authors
This manuscript entitled “Integrative Methylome and Transcriptome Analysis Reveals Epigenetic Regulation of Pigmentation in Oujiang Color Carp” presents an integrative analysis of MBD-seq and RNA-seq data across four pigmentation morphs of Oujiang color carp. The manuscript contains valuable data and has potential for publication in IJMS. However, I believe this manuscript requires major revision.
L17–18: It should be written as “(WR, WW), approximately 6% higher on average, …”.
L23: “gene” should be corrected to “genes”.
L147: What is the reference standard for the 96 pigmentation-related genes? Is it based on literature?
L237: It should be expressed as “it can interfere”.
L368–369: The DESeq2 threshold of |log2FC| ≥ 0.75 is slightly lower than the commonly used standard; please explain the reason.
Section 3.3: It is recommended to add a discussion on pigmentation and lipid metabolism in other closely related fish species.
The authors should better articulate the practical implications of their findings, e.g., potential use of methylation markers in selective breeding of ornamental or aquaculture carp.
Please unify terminology: “non-black spotted” or “non-black-spotted”
Ensure consistent figure citation format (e.g., “Fig. 1a–d” or “Figure 1a–d”).
Please ensure that Latin names in the references are italicized, and unify the reference format. For example, some references use “et al.” while others list all authors.
Reviewer 3 Report
Comments and Suggestions for Authors
This paper, using integrated genome-wide DNA methylation (MBD-seq) and transcriptome (RNA-seq) analyses, reveals the epigenetic regulatory mechanisms underlying color diversity in Oujiang colored carp (Cyprinus carpio var. color). The study found that black-spotted groups (RB, WB) had higher overall DNA methylation levels than unspotted groups (WR, WW), with methylation differences primarily located in intronic and intergenic regions. Through integrated analysis, the authors found a negative correlation between promoter methylation and gene expression and identified key genes, such as ASIP and frmA, that may regulate black spot formation through methylation-mediated silencing. Furthermore, pigmentation and lipid metabolism pathways were significantly enriched, suggesting a crucial role for epigenetic regulation in pigmentation and color stability. The paper innovatively integrated methylome and transcriptome data to systematically reveal the regulatory role of DNA methylation in pigmentation, particularly the negative correlation between promoter regions and gene expression. This multi-omics integration strategy is relatively uncommon in fish pigmentation research and provides new mechanistic insights. The study not only focused on pigment-related pathways (such as tyrosine metabolism and melanin synthesis) but also found significant enrichment in lipid metabolism pathways in red and white populations, suggesting that color formation may be related to energy metabolism and carotenoid transport, furthering our understanding of the mechanisms of fish color formation.
Several minor points could be improved.
1. Lack of functional validation experiments. While the paper proposes a regulatory model for genes such as ASIP and frmA, no functional experiments (such as CRISPR/Cas9, methylation inhibitor treatment, or transgenic validation) were performed to confirm causality. This weakens the reliability of the conclusions and is the primary limitation.
2. Small sample size and insufficient biological replicates. Only three individuals were used for each color group, all from the same full-sib family, which may not fully represent population diversity, limiting statistical power and the generalizability of the results.
3. Limitations of the methylation detection method. Using MBD-seq rather than whole-genome methylation sequencing (WGBS) may miss some non-CpG islands or hypomethylated regions, particularly regulatory elements outside of promoters, compromising the interpretation of the methylation landscape.
Round 2
Reviewer 1 Report
Comments and Suggestions for Authors
The study design is inadequate as it does not include true biological replicates. Observations made on a single individual cannot distinguish experimental effects from inherent biological variation. To draw meaningful conclusions, the experimental groups must include multiple biologically independent samples.
Author Response
Response to Reviewer 1 Comments
We sincerely appreciate your careful review and constructive feedback. We have responded your comment, we hope our answer can address your concerns and look forward to your evaluation.
Comments 1: The study design is inadequate as it does not include true biological replicates. Observations made on a single individual cannot distinguish experimental effects from inherent biological variation. To draw meaningful conclusions, the experimental groups must include multiple biologically independent samples.
Response 1: Thank you for highlighting the issue of experimental design. We clarify that each color group included three biologically independent individuals (n = 3), all drawn from the same full-sib family but sampled independently. We acknowledge that n = 3 per group imposes constraints on statistical power. We have therefore reported results with appropriate stringency and will expand sample sizes in subsequent studies to further strengthen the conclusions. In addition, RT-qPCR experiments validated our experimental design and findings, enhancing the credibility of our conclusions.
Reviewer 2 Report
Comments and Suggestions for Authors
Please carefully check the References: all Latin binomial names (genus + species) in the reference list should be italicized. such as Lates calcarifer(L595), Cypriniformes, Cyprinidae(L610).
Author Response
Response to Reviewer 2 Comments
Thank you for your positive assessment and helpful suggestions. We have corrected the reference formatting accordingly and look forward to your feedback.
Comments 1: Please carefully check the References: all Latin binomial names (genus + species) in the reference list should be italicized. such as Lates calcarifer(L595), Cypriniformes, Cyprinidae(L610).
Response 1: Thank you for the helpful reminder. We have systematically reviewed the entire reference list and corrected the formatting of biological names.
Location in the revised manuscript:
Page 17, Paragraph 1, Lines 524,525,529
Page 18, Paragraph 1, Lines 541, 544, 547, 564, 565
Page 19, Paragraph 1, Lines 611
Reviewer 3 Report
Comments and Suggestions for Authors
The authors have responded well to previous comments. I have no further questions.
Author Response
Response to Reviewer 3 Comments
Thank you for your positive evaluation. We are grateful that you find the English acceptable and that our previous revisions have addressed the concerns. We have carefully proofread the manuscript once more and confirmed that all prior changes remain accurately reflected in the latest files. We have no additional changes to propose at this time.
Comments 1: The authors have responded well to previous comments. I have no further questions.
Response 1: We thank the reviewer for the positive assessment and for the time devoted to evaluating our revision. We have re-checked the manuscript to ensure that all prior changes are correctly reflected and that the language remains clear. We have no further changes to propose at this stage.
Round 3
Reviewer 2 Report
Comments and Suggestions for Authors
There are no other problems with this paper
Author Response
Thank you for the positive evaluation. We appreciate your time and consideration.